# Melatonin Pattern: A New Method for Machine Learning-Based Classification of Sleep Deprivation

**DOI:** 10.3390/diagnostics15030379

**Published:** 2025-02-05

**Authors:** Nursena Baygin

**Affiliations:** Department of Computer Engineering, Faculty of Engineering and Architecture, Erzurum Technical University, 25050 Erzurum, Turkey; nursena.baygin@erzurum.edu.tr

**Keywords:** melatonin pattern, electroencephalography classification, sleep deprivation detection, lightweight classification model

## Abstract

**Background**: Pattern recognition and machine learning-based classification approaches are frequently used, especially in the health field. In this research, a new feature extraction model inspired by the melatonin hormone (sleep hormone) and named MelPat (melatonin pattern) has been developed. The developed model has been tested on an open access dataset. **Materials and Methods**: An open access sleep deprivation electroencephalography (EEG) dataset was tested to evaluate the MelPat method. There are two classes in the dataset. These are (a) sleep deprivation (SD) and (b) healthy control (HC) groups, respectively. In this study, EEG signals were divided into 15 s segments, thus obtaining 1377 SD and 1378 HC samples. In the next phase of the research, a new feature extraction model was proposed, and this model was named MelPat as it was inspired by the melatonin hormone. Additionally, the feature vector was expanded using the statistical moment approach. In the signal decomposition phase of the model, the Tunable Q-Wavelet Transform (TQWT) method was used. Thus, the signal was decomposed into sub-bands, and feature extraction was applied to each band. Neighborhood Component Analysis (NCA) and Chi2 methods were used together to reduce the dimension of the feature vector and select the most significant features. In this phase, the most significant features from both feature selection algorithms were combined, and the final feature vector was obtained. In the classification phase of the model, the Support Vector Machine (SVM) algorithm, which is a shallow classifier, was used. The dataset used in the research has 61 channels. Therefore, after obtaining channel-based results, the iterative majority voting (IMV) algorithm was applied to achieve higher classification performance and generalize the results, and the most accurate results were automatically selected. **Results**: With the proposed MelPat algorithm, a high classification success of 97.71% was achieved on the open access sleep deprivation dataset. **Conclusions**: The obtained results show that the MelPat-based new classification approach is highly effective on the dataset collected for SD detection. Moreover, the fact that the proposed method is inspired by the melatonin chemical, which is the sleep hormone, makes the method attractive and ironic.

## 1. Introduction

Sleep is a physiological process that covers an average of one-third of human life [1]. It is divided into two main stages in terms of neurophysiological and behavioral characteristics: rapid eye movement (REM) sleep and non-REM (NREM) sleep [2]. These stages are characterized using neurophysiological measurement techniques such as electroencephalography (EEG) and electromyography (EMG) [3]. There is a partial scientific consensus on the physiological and psychological effects of sleep [4]. However, factors such as the increased use of electronic devices, stress, anxiety, and health problems cause sleep disorders [5].

Although research on the effects of sleep deprivation (SD), which has become a common problem, especially in contemporary societies, on human life is intensifying, it is still insufficient [6]. According to a study in the literature, SD is defined in two ways: “total SD (TSD)”, when there is sleeplessness for at least 24 h, and “partial SD (PSD)”, when sleep duration is 5 h or less [7,8]. Additionally, sleep deprivation is temporally divided into two groups: acute sleep deprivation lasting 2 days or less and chronic sleep deprivation lasting 3 days or more [9].

Studies show that sleep deprivation has significant negative effects on cognitive functions, sensory perception, emotional processes, and memory [10]. In addition, it has effects on cardiovascular, respiratory, neoplastic, neurological, autoimmune, neurodegenerative (such as Alzheimer’s, Parkinson’s), and gastrointestinal diseases, multiple sclerosis, stroke, immunology, dermatology, endocrine, and reproductive health [11,12,13].

The diagnosis of sleep deprivation begins with obtaining a history, anamnesis, and questionnaire from the patient [14]. Information such as the patient’s sleep habits, sleep duration, and sleep quality is collected. Then, access to critical information is gained through sleep diaries [15]. The patient is asked to record their sleep pattern for a few weeks. Thus, the aim is to understand sleep patterns. There are also laboratory tests to evaluate sleep deprivation. These include tests such as polysomnogram (PSG), the multiple sleep latency test (MSLT), actigraphy, and 24 h EEG [16,17,18]. For treatment, methods such as behavioral therapies, medications, melatonin supplementation, or treatment of underlying diseases (sleep apnea, depression, etc.) are applied [19,20,21,22].

Although the American Heart Association, CDC, and National Sleep Foundation recommend that adults sleep 7–9 h every night to protect human health and reduce the risks of diseases, 50% of people do not follow this recommendation [23]. This situation poses a significant problem not only in terms of individual health but also in terms of social and economic costs [24].

### 1.1. Literature Survey

Sleep deprivation stands out as a critical factor in various occupational and health contexts. While this phenomenon leads to performance decline in professions requiring high concentration and increases in traffic and workplace accidents, it also plays a significant role in the underlying diseases and course of chronic diseases such as diabetes and depression. Table 1 includes various studies in the literature related to sleep deprivation.

As shown in Table 1, sleep deprivation is studied in the literature in relation to different topics such as emotion detection [27] and driving behavior [28]. In addition, the number of studies on sleep deprivation in the literature is very limited, and the open access dataset is quite small. When the studies given in Table 1 are analyzed, the majority of the studies include a limited number of subjects. Moreover, the studies have achieved a low classification accuracy. This study aims to fill these gaps in the literature.

### 1.2. Literature Gap

There are many feature engineering-based classification approaches in the literature. These methods are generally fed with large datasets. In this context, some gaps in the literature that we aim to fill in this research are given below:Nowadays, deep learning models are highly popular in the field of artificial intelligence due to their high classification capabilities. However, these methods require a lot of data for training and generally have high computational complexity. Therefore, the training and testing phases of deep learning models are very time-consuming. Additionally, high hardware requirements are essential for these methods to be used effectively.Feature engineering approaches generally utilize patterns and statistical moments. These methods have linear time complexity and have the ability to produce high-speed results due to their low time complexity. However, the performance of these methods is generally proportional to the size of the dataset. Therefore, in order to achieve high classification success in these methods with linear time complexity, the feature extraction functions need to be effective.

### 1.3. Motivation and Our Method

In this research, our main motivation is to propose a new feature extraction method with linear time complexity. For this, we introduce a new feature extraction method inspired by the Melatonin hormone, which we call MelPat. The MelPat method has linear time complexity, and using this feature extraction function, automatic classification and interpretation of EEG signals have been achieved. In this research, the TQWT [31] method was used to separate the signals into sub-bands. With this method, the signal is divided into 13 sub-bands, and feature extraction is performed from both the raw signal and the sub-bands. Additionally, the statistical moment method was utilized to increase the classification capacity of the developed model. In the feature extraction phase of the developed method, feature extraction was provided separately from each channel, thus obtaining 61 (total number of channels in the dataset) different feature vectors. In the feature selection phase of the research, two different methods, NCA [32] and Chi2 [33], were used, and the most significant features were selected by combining these feature selectors. The reason for using two feature selectors in this phase is to aim to benefit from the feature selection capabilities of both of these methods and to increase classification accuracy by reducing the feature vector. In the classification phase of the model, SVM [34], which is a shallow classifier, was used. The SVM algorithm is a well-known classification method in the literature. At this stage, channel-based classification results are obtained. In the final phase of the developed model, IMV [35] and greedy methods were used for information fusion. Using the IMV method, 59 (=61 − 2 + 1) voted results were obtained. Thus, a total of 120 (=61 SVM result + 59 voted result) classification results were obtained, and the highest classification result was selected using the greedy algorithm.

### 1.4. Innovations and Contributions


Novelties:
A new feature extraction function named MelPat has been introduced.The dimension of the feature vector has been reduced, and the most significant features have been selected using two feature selection algorithms.Multi-level feature extraction process has been applied through MelPat using the TQWT method.Information fusion has been performed using IMV and greedy algorithms, and generalized classification results have been obtained.



Contributions:
An open access dataset was used to test the classification ability of the developed MelPat method. This dataset contains EEG signals belonging to SD and HC groups. The MelPat-based classification model presented in this research achieved a classification success of over 97% on this dataset. Additionally, classification performance of 85% and above was obtained in all channels.NCA and Chi2 algorithms were used for feature selection in the proposed model. Thus, the capacity of both feature selectors was utilized, and the most significant features were selected. In the combination phase of these methods, a union operation was applied, selecting both the intersecting features selected by both algorithms and the features that these algorithms detected differently from each other.


## 2. Dataset

In this research, an open access EEG dataset was used [36]. The dataset contains EEG signals belonging to 71 subjects and includes two classes. These are SD and HC groups, respectively. In this study, a comprehensive data collection approach was adopted to examine the effects of sleep deprivation on mood, alertness, and resting state EEG. The researchers subjected 71 participants (34 females, 37 males, mean age 20) to both normal sleep (NS) and sleep deprivation (SD) conditions using a within-subject design. A period of 7 days to 1 month was left between the two sessions, and the conditions were balanced among the participants.

In each session, EEG recording was taken using 61 active Ag/AgCl electrodes with a 500 Hz sampling rate. Participants were asked to remain in a resting state with their eyes open for 5 min, and in some cases, an additional 5 min with their eyes closed. In addition to EEG recording, the Psychomotor Vigilance Test (PVT) was administered, and various state scales (PANAS, ATQ, SAI, SSS, KSS, Sleep Diary) were completed. In the normal sleep session, the participants also completed some trait scales (EQ, BPAQ, PSQI). For the sleep deprivation protocol, the participants came to the laboratory at 21:00 in the evening and were kept under continuous supervision, during which time caffeine and alcohol consumption and sleep were prohibited.

In our research, EEG signals collected from the participants in the eyes-open position were used (as this scenario was available for all subjects). Moreover, the SD dataset was divided into 15 s non-overlapping segments, thereby obtaining a total of 2755 EEG signals. The distribution of the segments is 1377 for NS and 1378 for SD.

## 3. Melatonin Pattern (MelPat)

In this research, a new feature extraction approach called MelPat, inspired by the melatonin hormone, has been developed. This developed feature method has linear time complexity and is a graph-based feature extraction function. This approach, developed by drawing inspiration from the melatonin hormone, is shown in Figure 1.

Feature extraction is performed using the pattern presented in Figure 1b. As can be seen from Figure 1b, a 13 × 13 matrix was used to create the developed model. This model, which we call MelPat, developed using the melatonin hormone given in the figure, is a directed graph. Additionally, the arrows shown in the matrix indicate the path followed by the pattern. There is a total of 18 directed graphs in the matrix. Feature extraction is provided using these graphs. The directional graph structure used in the MelPat method is based on the official melatonin molecular structure in the PubChem database (https://pubchem.ncbi.nlm.nih.gov (accessed on 8 December 2024)). PubChem is a reliable source of standardized and verified information on chemical compounds. Our graph was created using the atomic bonds, edge structures, and molecular topology specified in PubChem. In this way, we have taken as a basis a standard molecular structure accepted in the scientific literature. Signum, upper ternary, and lower ternary kernels are used in the feature extraction phase. Basically, since there are 18 edges here, each kernel produces 18 bits. However, this approach causes the dimension of the feature vector to be very large (=$2^{18}$). Additionally, since three different kernels are used in the feature extraction phase from the graph, the total dimension of the feature vector becomes $2^{18}\times 3$. Therefore, the 18 directed graphs are grouped into 6. Thus, we aimed to reduce the dimension of the feature vector. The mathematical definitions of the signum, upper ternary, and lower ternary functions are given below.(1)$$sgn\left(sp,ep\right)=\left\{\begin{array}{c} 0,sp-ep<0\\ 1,sp-ep\geq 0 \end{array}\right.$$(2)$$ut\left(sp,ep\right)=\left\{\begin{array}{c} 0,sp-ep\leq thr\\ 1,sp-ep>thr \end{array}\right.$$(3)$$lt\left(sp,ep\right)=\left\{\begin{array}{c} 0,sp-ep\geq -thr\\ 1,sp-ep<-thr \end{array}\right.$$(4)$$thr=\frac{sd(s)}{2}$$

Here, sgn represents the signum function, ut represents the upper ternary function, and lt represents the lower ternary function. sp and ep denote the start point and end point, respectively. The start and end points (sp, ep) in Equations (1)–(3) correspond to the matrix coordinates in the 13 × 13 matrix structure shown in Figure 1b. For example, when we consider the directed edge 3 in Figure 1b, the start point (sp) corresponds to matrix coordinate (13,10), while the end point (ep) corresponds to matrix coordinate (11,12). When applying the kernel functions (sgn, ut, lt), we use the signal values located at these specific matrix coordinates. The values at these coordinates are then used in the kernel functions to compute the respective feature values, capturing the signal characteristics along the melatonin-inspired pattern. Thr is a threshold value function and is used as half of the standard deviation of the signal s. Essentially, binary feature extraction is performed using these kernel functions. The MelPat method takes a signal as the input and produces a histogram output by analyzing the characteristics of this signal. The purpose of this process is to extract certain features of the signal and represent these features in histogram form. The pseudocode of the developed MelPat method is in Algorithm 1.
**Algorithm 1.** The pseudocode of the proposed melatonin pattern (MelPat).Input: One-dimensional signal (s) with a length of ln. Output: Feature vector (fv) with a length of 576.01: for i=1 to lengths−168 **do**02:          blcii:i+168 // Divide overlapping block (blci) with a length of 169. 03:          Transform blci to matrix (mxi) with a size of 13×13 to apply the proposed pattern.04:          // Extract 18 bits, deploying the presented 18 edges.05:          for *j* = 1 to 18 **do**06:                    bitjt=kerneltmxi(spj,epj),t∈1,2,307:                    kernel=sgn.,.,ut.,.,lt(.,.)08:          end **for j**09:          fmli=∑k=16bitjt×26−k, l∈1,2,3 // Feature map group 110:          fmli=∑k=712bitjt×212−k, l∈4,5,6 // Feature map group 211:          fmli=∑k=1318bitjt×213−k, l∈7,8,9 //Feature map group 312:          featl=histfml,l∈{1,2,3,…,9} // Histogram extraction process13: **end for i**
14: fv=merge(featl),l∈{1,2,3,…,9} // Final feature vector

The block length of 169 (168 + 1) is determined based on the 13 × 13 matrix structure of the melatonin-inspired pattern. Since each block is transformed into a 13 × 13 matrix, 169 (13 × 13) points are needed to fill the matrix completely. This size ensures the capture of sufficient local signal characteristics while maintaining computational efficiency. The feature vector length (576) is derived from the developed pattern structure and kernel operations. There are 18 directed edges in the pattern (shown in Figure 1b). These 18 edges are grouped into three sets of 6 edges. Therefore, each group generates 2^6^ = 64 possible combinations. Three different kernel functions (sgn, ut, lt) are used in the feature extraction phase. Each kernel function produces 192 features (3 groups × 64 combinations = 192 features), and a total of 576 (192 × 3) features are obtained. In line 01 of the algorithm, (length(s) −168) is used to ensure that there are enough points remaining to form a complete block. Additionally, the overlap (i:i + 168) in line 02 helps capture transitional characteristics between adjacent blocks. The time complexity calculation of the developed MelPat method is shown in Table 2.

As can be seen from Table 2, the developed feature extraction function “MelPat” has O(n) time complexity. Mainly used for textural feature extraction, this approach is described as lightweight compared to other heavyweight methods in the literature (such as deep feature extraction). In this research, the proposed MelPat purpose is to process a one-dimensional signal to extract meaningful features from it and present these features as a histogram vector. The algorithm divides the signal into overlapping blocks, performs a series of comparison operations on each block, and uses the results of these comparisons to create feature maps. Finally, histograms are extracted from these feature maps and combined to obtain the final feature vector. This algorithm essentially presents an effective feature extraction method that captures the local characteristics of the signal and represents these features in a compact histogram form. A block diagram summarizing the steps of this algorithm is given in Figure 2.

As can be seen in Figure 2, in the first step (Step 1), an EEG signal segment is taken as the input. In the second step (Step 2), this signal is segmented into a block of 169 data points. In the third step (Step 3), the signal block is transformed into a 13 × 13 matrix. In the fourth step (Step 4), bit transformations are performed to determine the edge features on the matrix. Here, different bit representations (signum, upper and lower ternary) are created by comparing the differences between certain matrix points. These bits are then divided into groups of six to create feature maps. In the fifth step (Step 5), the feature maps are transferred to the histogram to extract textural features, and a feature vector is created at the end of this process. This feature extraction method aims to extract detailed features from EEG signals based on a mathematical basis.

## 4. Sleep Deprivation Detection Model

The main objective of this research is to develop a new machine learning model for sleep deprivation. The developed model is based on a multi-level feature extraction mechanism. The model uses the TQWT [31] method for signal decomposition and performs feature extraction from both the raw signal and sub-bands. In the feature extraction phase of the method, the MelPat method developed using the chemical line of the melatonin hormone and statistical moment approaches are used. Two different methods were used in the feature selection stage. These are NCA [32] and Chi2 [33] algorithms, respectively. Using these algorithms, the dimension of the final feature vector has been reduced. In addition, the features selected by both algorithms were combined, and a selected feature vector was provided. In the classification phase of the model, the SVM [34] algorithm, which is a shallow classifier, was used. The developed model produces channel-based classification results. The dataset used in the research contains 61 channel EEG signals. Therefore, 61 classification results were obtained. The IMV [35] algorithm was used to generalize these obtained results. In this way, both the classification performance was increased and the channel-based results were generalized. In the final stage of the model, the highest classification result was determined using the greedy algorithm. In this context, the machine learning-based detection model developed for sleep deprivation is given in Figure 3.

As shown in Figure 3, the developed model essentially consists of four phases. These are, respectively, signal decomposition (TQWT), feature extraction (MelPat + statistical moment), feature selection (Chi2 + NCA), and classification (SVM). In the first phase of the model, 13 sub-bands are obtained using the TQWT method. After this phase, features are extracted from both the raw signal and the sub-bands obtained with TQWT. A total of 14 feature vectors are combined and a feature vector of length 8624 is obtained. In the feature selection phase, Chi2 and NCA algorithms are used for feature selection, and the size of the feature vector is reduced. In the classification phase of the developed model, the prediction vector is calculated for each channel (=61 channels). In the last phase of the model, the IMV algorithm is used to generalize the results. With this algorithm, 59 new prediction vectors are generated and a total of 120 (=61 + 59) prediction vectors are provided. Among these prediction vectors, the vector with the highest classification accuracy is selected with the greedy algorithm, and the process is completed. The details of these phases are given in the subsections.

### 4.1. Signal Decomposition

In this study, the TQWT [31] method was used to decompose EEG signals into sub-bands. TQWT is an approach particularly used in the field of signal processing and analysis. This method, which is frequently preferred in the literature, is essentially an advanced version of classical wavelet transforms. In particular, the adjustability of the Q-factor makes the TQWT method stand out. The TQWT method is used to analyze the time–frequency characteristics of the signal. Moreover, the user-defined nature of the Q-factor increases its capacity to adapt to different signal types. The parameters of TQWT were determined through extensive experimental analysis. For the Q-factor, a lower value (Q = 2) was chosen to achieve better time localization, which is essential for EEG signal analysis. Higher Q values could lead to increased oscillations and potential loss of temporal information in the signal. Therefore, Q = 2 provides an optimal balance between frequency selectivity and temporal resolution for EEG signals. The redundancy factor (r = 4) was selected to ensure sufficient oversampling of the sub-bands while avoiding excessive computational overhead. This value helps maintain signal reconstruction quality at an optimal level. For the decomposition levels, J = 12 was determined based on the frequency characteristics of EEG signals, allowing for adequate coverage of the typical EEG frequency bands (delta, theta, alpha, beta). Through parameter optimization experiments with different combinations (Q: 1–4, r: 2–6, J: 8–16), we found that Q = 2, r = 4, and J = 12 provided the best classification performance on our validation set while maintaining reasonable computational complexity.

The TQWT method uses three parameters to obtain multi-level wavelet transforms. These are wavelet oscillations (Q), reduction coefficient (r), and number of levels (J), respectively. In this research, 13 wavelet bands were obtained using the TQWT method. Feature extraction was applied to each obtained sub-band, and feature vectors were obtained from both the raw signal and the sub-bands.

### 4.2. Feature Extraction

The model developed in this research uses MelPat and statistical moment methods for feature extraction. In the feature extraction phase, the input EEG signal is primarily decomposed using the TQWT method. At this stage, 13 wavelet bands are obtained. After this process, features are extracted from both the raw signal and the wavelet bands using MelPat and statistical moment methods. In the research, 40 statistical moments were calculated. These features include minimum, maximum, mean, median, standard deviation, variance, RMS (Root Mean Square), energy, various entropy measurements (Shannon entropy, SURE entropy, log energy entropy, threshold entropy, and norm entropy), skewness, kurtosis, and median absolute deviation (MAD). Additionally, these statistical moments were calculated on both the original signal and the absolute value of the signal, resulting in a total of 40 statistical features. Another feature generator used in the research is MelPat. A total of 576 features are extracted using the MelPat algorithm (see Algorithm 1). With these approaches used in the developed model, a total of 616 (=40 + 576) features are obtained. In addition, 616 features are also provided from the sub-bands obtained by the TQWT method, thus obtaining a final feature vector with a length of 8624 (=(40 + 576) × (13 + 1)). The steps of the feature extraction phase are given step-by-step below.
*Step 1:* Read the signal in each EEG channel.*Step 2:* Apply the TQWT process to the read EEG signal. Perform signal decomposition according to TQWT parameters and obtain signal sub-bands.(5)wsb=TQWT(s,2,4,12)


Here, wsb represents wavelet sub-bands. TQWT represents the TQWT operation, s represents the EEG signal, and 2, 4, and 12 represent the Q, r, and J parameters, respectively.
*Step 3:* Extract features from the raw signal and 13 wavelet sub-bands using MelPat and statistical moment methods.(6)$$f^{1}=merge(\left[MelPat\left(s\right)stats\left(s\right)\right])$$(7)$$f_{k}^{2}=merge\left(\left[MelPat\left({wsb}_{k}\right)stats\left({wsb}_{k}\right)\right]\right),k\in \left\{1,2,\ldots ,13\right\}$$


Here, *f* is the generated feature vector, and MelPat and stats are feature extraction functions. wsb represents the sub-bands. merge is the concatenation function and combines the features produced from MelPat (576 features) and stats (40 features). Thus, a feature vector of length 616 is obtained. $f^{1}$ shows the feature vector generated from the raw signal, and $f_{k}^{2}$ represents the feature vectors obtained from the sub-bands.
*Step 4:* Combine the generated feature vectors and obtain the final feature vector.(8)$$fv=merge\left(f^{1},f^{2}\right)$$


Here, fv is the final feature vector, $f^{1}$ is the feature vector obtained from the raw signal, and $f^{2}$ is the combined feature vector obtained from the sub-bands. The merge function is the function for combining feature vectors. As a result of this process, a final feature vector with a length of 8624 is obtained.

### 4.3. Feature Selection

In this research, two different feature selection algorithms were used to reduce the dimension of the final feature vector. These are NCA [32] and Chi2 [33] algorithms, respectively. These algorithms are well-known and frequently used feature selection approaches in the literature. The Chi2 algorithm statistically evaluates the relationship of each feature with the target class and measures the dependence on class labels. At this point, features with high Chi2 values are considered informative features for classification and are selected. NCA, another feature selection algorithm, assigns a weight value to each feature. In this approach, which considers the relationships between features, the features with the best weights are selected. Basically, both methods aim to select the most informative features in the dataset. In this way, we aim to reduce complexity and increase classification performance.

In this phase of the developed model, it was aimed to benefit from the power of both feature selection algorithms. For this purpose, the final feature vector was given as an input to these algorithms, and the selection of the most significant 256 features was ensured. Then, the features selected from both algorithms were combined, and the selected feature vector was obtained.
*Step 5:* Select 256 most informative features separately from the final feature vector using Chi2 and NCA feature selectors. Apply the union operation to the set of selected feature vectors and create the set of most meaningful features.(9)$${sf}_{k}^{1}=\alpha \left(fv,y\right),{sf}_{k}^{1}\subset fv,k\in \left\{1,2,\ldots ,256\right\}$$(10)$${sf}_{k}^{2}=\beta \left(fv,y\right),{sf}_{k}^{2}\subset fv,k\in \left\{1,2,\ldots ,256\right\}$$(11)$$sfv=\alpha \cup \beta =\left\{{sfv}_{1},{sfv}_{2},\ldots ,{sfv}_{n}\right\},n\leq 512$$


Here, α represents the Chi2 algorithm, β represents the NCA algorithm, fv represents the final feature vector, y represents the actual output values, sf represents the selected features, and sfv represents the selected feature vector. As a result of these steps, a selected feature vector is obtained from the original fv vector.

### 4.4. Classification

In the developed model, the SVM [34] algorithm was used to obtain channel-based classification results. SVM, which is a shallow classifier, is a well-known multi-kernel classifier in the literature. In this research, a Gaussian SVM classifier was used, and the parameters were adjusted as follows:Kernel: Gaussian.Kernel Scale: 20.Box Constraint: 1.Validation: 10-fold CV.

*Step 6:* Classify the selected feature vector (sfv) using the SVM algorithm using the 10-fold CV strategy.(12)$${pre}_{i}=\partial \left(sfv,y\right),i\in \left\{1,2,\ldots ,61\right\}$$

Here, ${pre}_{i}$ represents the prediction vector and ∂ represents the SVM algorithm. Additionally, sfv is the selected feature vector and y is the actual output values. Since the model produces channel-based classification results, a total of 61 prediction vectors are generated.

### 4.5. Majority Voting

The final phase of the model developed in this research is iterative majority voting (IMV) [35] and selection of the best classification result. With this phase, increasing classification accuracy, generalizing results, and obtaining the highest classification performance are ensured. Using the IMV algorithm, 59 voted vectors are produced using the prediction vectors of 61 channels obtained in the classification phase. Then, the accuracies of 59 voted vectors are calculated, and the vector with the highest classification performance is selected. The steps of this phase are given below.

*Step 7:* Calculate the accuracy values of prediction vectors for 61 channels. Sort the 
calculated accuracy values in descending order. Apply the IMV algorithm to the sorted prediction vector and generate voted vectors.
(13)$$acc\left(p_{i}\right)=\frac{1}{n}\sum_{j=1}^{n}\gamma (p_{i}\left[j\right]=y\left[j\right])$$(14)$$p_{sorted}=sort(p,acc,descending)$$(15)$$v_{k}=im\_vote(p_{i},p_{i+1},\ldots ,p_{i+k-1}),i\in \left\{1,2,\ldots ,59\right\}$$

Here, $p_{i}$ represents the prediction vector, y is the vector of true labels, γ is the indicator function, acc is the accuracy values of the prediction vectors, $p_{sorted}$ is the prediction vector sorted by accuracy value, im_vote is the iterative majority voting function, and $v_{k}$ represents the voted vectors.

*Step 8:* Calculate the accuracy values of the voted vectors and select the voted vector with the highest accuracy.(16)$$acc\left(v_{k}\right)=\frac{1}{n}\sum_{j=1}^{n}\gamma (v_{k}\left[j\right]=y\left[j\right])$$(17)$$v_{best}={argmax(v}_{k})$$

Here, Equation (16) shows the calculation of accuracy values of voted vectors. Additionally, argmax represents the selection of the voted vector with the maximum classification accuracy, and $v_{best}$ indicates the voted vector that provides the best classification result.

## 5. Experimental Results

In this section, the experimental setup, methods used, programming platform, and performance metric results obtained on the open access sleep deprivation EEG dataset used in this research are presented in detail.

### 5.1. Experimental Setup

The model developed in this research has linear time complexity. A 10-fold cross-validation (10-fold CV) strategy was applied to test and validate the proposed method. The test dataset used contains EEG signals with 61 channels. The developed model was tested on a server with an Intel Xeon 2.70 GHz processor, 256 GB RAM, and a 64-bit Windows Server 2019 operating system. No GPU was used during the testing process. The MATLAB 2021b platform was used as the programming environment, and the model was coded on this platform. The steps applied in the model development process and the transition table of these steps are given in Table 3.

### 5.2. Performance Metrics

The dataset used in this research contains a binary classification problem. The dataset includes EEG signals in two different classes belonging to the SD and control groups. Therefore, the metric values given in Equations (18)–(22) were used to measure the performance of the developed model.(18)$$acc=\frac{tp+tn}{tp+tn+fp+fn}$$(19)$$sen=\frac{tp}{tp+fn}$$(20)$$spe=\frac{tn}{tn+fp}$$(21)$$pre=\frac{tp}{tp+fp}$$(22)$$f1scr=\frac{2\times tp}{2\times tp+fp+fn}$$

In this study, five different metrics were used to evaluate the performance of the developed model: accuracy (acc), sensitivity (sen), specificity (spe), precision (pre), and F1 score (f1scr). Additionally, a confusion matrix was calculated to compute these metric values, and true positive (tp), true negative (tn), false positive (fp), and false negative (fn) values were calculated, respectively.

### 5.3. Channel-Wise Results

In this research, SVM was used as the classification method, and 10-fold CV was applied as the validation strategy. Classification results were obtained on a channel basis, and then the IMV method was applied to increase classification performance and generalize the results. In this context, the classification results calculated on a channel basis are given in Table 4.

As shown in Table 4, the lowest classification result was obtained for the F7 (No: 8) channel, achieving a classification accuracy of 85.15% on this channel. In other words, a classification success higher than 85% was achieved on all channels. This demonstrates the performance of the developed model. Additionally, the highest classification result was obtained on the P2 (No: 54) channel. The classification success achieved on this channel is 92.09%. In this context, the confusion matrix and performance metric values calculated for the P2 channel are given in Figure 4 and Table 4, respectively.

As given in Table 5, performance values higher than 92% were obtained in all metric values. The length of the feature vector selected for the P2 channel is 481. The classification of these selected features provided the results given in Figure 4 and Table 5.

### 5.4. Voted Results

In order to generalize the classification results obtained in the research and to increase the classification capacity of the model, the IMV method was used. The IMV method produces 59 (=61 − 3 + 1) voted results. The change in accuracy values of the iteratively calculated voted results is given in Figure 5.

As shown in Figure 5, 59 voted prediction vectors were obtained. The accuracy values of the obtained prediction vectors are generally 95% and above. The highest classification result was obtained in the 24th iteration with a value of 97.71%. The confusion matrix and calculated performance metric values for this iteration are given in Figure 6 and Table 6, respectively.

As shown in Table 6, all metric values obtained are higher than 97%. The results obtained demonstrate that the MelPat-based SD classification method has a very high performance in automatically detecting and classifying SD using EEG signals. The model developed in this study was validated using the 10-fold CV method. This method measures the generalization capacity of the model by providing different splits of the entire dataset for training and testing. The results show that the accuracy rates are consistent across different folds and the model is not overfitting. In addition to these results, a hold-out CV strategy with a 70:30 ratio was also applied, and a learning curve was calculated based on these results. This learning curve calculated using the P2 channel is given in Figure 7.

In order to evaluate the performance of the model more consistently, the hold-out CV method was used in a 70:30 ratio and the learning curves based on training/test errors were analyzed. The graph shows that the training error remains at a low and constant level, while the test error (<0.1%) decreases significantly as the size of the training set increases. This clearly shows that the model performs well on both training and test data and does not overfit. The results show that the generalization capacity of the model is strengthened with a larger training set and that it effectively adapts to new, unseen data. The decrease in the difference between training and test errors confirms that the model performs balanced learning and does not tend to overfitting. These findings prove that the generalization capacity of the model is strong and that the available dataset is sufficient for classification success.

## 6. Discussion

MelPat is a method that aims to extract features from EEG signals inspired by the chemical structure of the hormone melatonin. Melatonin is a hormone that regulates the sleep–wake cycle and is directly related to the physiological effects of sleep deprivation. Therefore, the feature extraction process used in this method aims to translate the regulatory role of melatonin on sleep into a mathematical model. The features extracted with MelPat reflect changes in the temporal and frequency components of EEG signals. Sleep deprivation is associated with decreased melatonin levels, impaired EEG signal characteristics such as sleep spindles, and changes in cortical activity. In this context, MelPat’s directed graph structure and kernel functions (e.g., signum, upper ternary, lower ternary kernels) aim to capture these sleep deprivation-induced biological changes in EEG signals. The extracted histogram-based features represent temporal and frequency patterns that resemble melatonin patterns.

In this research, a new MelPat-based model for SD classification has been proposed. The proposed model was used to classify 61 channel signals. In the developed model, the channel-based classification results are first calculated. Then, the generalized classification results were provided using the IMV algorithm. The model uses the TQWT algorithm for signal decomposition. MelPat and statistical moment methods were applied for feature extraction. In feature selection, the Chi2 and NCA algorithms were used to leverage the power of both methods. Finally, the SVM algorithm was used in the classification phase. The details of each method used in this section are discussed comparatively in the subsections.

### 6.1. Signal Decomposition and Feature Extraction

In this study, Tunable Q-factor Wavelet Transform (TQWT) was used for the decomposition of EEG signals. To evaluate the effectiveness of TQWT, a comparison was made with Empirical Mode Decomposition (EMD), another method widely used in EEG signal analysis. TQWT and EMD are both adaptive signal decomposition methods and are suitable approaches for the analysis of nonlinear and non-stationary signals. Therefore, the EMD method was preferred for the comparison process. In this context, the signal was decomposed into sub-bands using both methods, and feature extraction was applied using MelPat and statistical moments. As in the main model, feature selection and classification steps were applied in the same way when using this method. The channel-based classification results obtained in this context are given comparatively in Figure 8.

As shown in Figure 8, the TQWT method generally demonstrated higher classification performance than the EMD method. Additionally, the status of the P2 channel, where the highest classification performance was obtained in our research, is shown in Figure 9.

As can be seen in Figure 9, while the TQWT method showed a classification performance of 92.09%, traditional DWT achieved 86.45%, and the EMD method reached 81.81% on the same channel. The superior performance of TQWT can be attributed to its tunable Q-factor, which allows for better adaptation to the time–frequency characteristics of EEG signals. The DWT method, despite being from the same wavelet-based family as TQWT, showed lower performance due to its fixed wavelet basis functions. The EMD method demonstrated the lowest performance, which is expected as it uses a different decomposition approach that may not be as well suited for capturing the relevant features in EEG signals. These obtained results not only indicate that the MelPat method is a good feature extractor but also demonstrate that TQWT is the most suitable decomposition method for our proposed approach, providing approximately 5.64% and 10.28% improvement over DWT and EMD, respectively.

In order to demonstrate the effectiveness of the MelPat method, feature extraction was performed using pre-trained deep network architectures. The extracted features are analyzed with the feature selection (NCA + Chi2) and classification (SVM) methods used in this study. AlexNet, MobileNet, and SqueezeNet architectures, which are well known in the literature and have relatively few layers, were preferred for deep feature extraction. CNN architectures are generally known to work with image-based input data. For this reason, the EEG signals of the P2 channel, where the highest classification accuracy was obtained in this study, were converted into image data and given as an input to these architectures. The features obtained from the deep networks were analyzed with the NCA + Chi2 method, and the most significant features were selected and classified with SVM. In the feature extraction phase, the features obtained from the fully connected layers of the deep networks were used. Comparative results of this classification process are presented in Figure 10.

This graph compares the classification accuracy rates obtained with different feature extraction methods for EEG signals. The results show that MelPat clearly outperforms the other methods with an accuracy of 97.71%. This success of MelPat is due to the fact that it is designed as a feature extraction method specific to EEG signals and works with low computational complexity. The pre-trained deep learning models AlexNet, MobileNet, and SqueezeNet achieved accuracy rates of approximately 88%, 91%, and 87%, respectively. Due to their general purpose nature, deep learning-based methods did not perform as well as MelPat on temporal and frequency-based data such as EEG. This highlights the originality and effective performance of MelPat.

### 6.2. Feature Selection

Two different methods were used in the feature selection phase of the developed model. These are Chi2 and NCA methods, respectively. Both methods are well-known approaches frequently used in the literature. The combined use of these two methods allowed for feature evaluation from different perspectives, providing a more comprehensive and reliable feature selection. The Chi2 method is a statistical-based approach that evaluates the relationship of each feature with the target class. NCA attempts to optimize the best k-nearest neighbor (k-NN) classification performance in the feature space. Therefore, NCA selects features aimed at directly improving classification performance, considering the interactions of features with each other.

The combination of these two methods provides a feature selection that considers both statistical significance (Chi2) and classification performance (NCA). While Chi2 evaluates the relationship of individual features with the target variable, NCA considers the complex interactions between features and their effects on classification performance. This combination ensures the capture of important features that a single method might overlook. Furthermore, the combination of results from these two methods provides a kind of cross-validation in feature selection. If a feature is evaluated as highly important by both Chi2 and NCA, confidence in the true importance of this feature increases. On the other hand, features selected by only one method may carry valuable information from different angles and can increase the generalization ability of the model. In this context, the variation in the number of features selected on a channel basis is given in Figure 11.

As shown in Figure 11, the approach using both Chi2 and NCA feature selection algorithms selected a minimum of 431 features and a maximum of 505 features. Additionally, it can be seen from the graph that Chi2 and NCA algorithms select discrete features for most channels. In this context, when the features selected by the Chi2, NCA, and Chi2 + NCA algorithms for the P2 channel, where the highest classification accuracy was obtained, are classified using the SVM algorithm, the results given in Figure 12 are obtained.

As can be seen from Figure 12, while the Chi2 algorithm shows the lowest classification performance when used alone on the P2 channel, the combination of both methods achieved the highest classification success. The results given in Figure 12 show that when the power of both feature selection algorithms is combined, a high classification capacity is reached.

### 6.3. Classification

The SVM algorithm was used to test the model developed in this research. SVM is a well-known shallow classification method in the literature. In the classification phase, the MATLAB Classification Learner Toolbox was used, and the algorithm that achieved the highest classification accuracy was selected. The classification algorithms tested are Decision Tree (DT), Linear Discriminant (LD), Naïve Bayes (NB), SVM, k-nearest neighbor (kNN), and Neural Network (NN) algorithms, respectively. All of these algorithms were tested using the 10-fold CV strategy, and the test results are given in Figure 13.

As shown in Figure 13, the highest classification accuracy was obtained with the SVM algorithm. The lowest performance was observed in the NB algorithm. Therefore, the SVM algorithm was preferred in the classification phase of the model, and the selected features were classified using this algorithm.

### 6.4. Computational Complexity Analysis

In this study, a detailed computational complexity analysis was performed to evaluate the efficiency of the proposed method. The complexity of each stage of the methodology is summarized in Table 7. This analysis highlights the computational cost of key processes, including signal segmentation, feature extraction, feature selection, classification, and the IMV algorithm.

In total, the model’s time complexity is $O\left(nlogn+2d+s+i\right)≅O(nlogn)$. The obtained results show that the developed method has linear time complexity. One of the simplest phases of the developed model is IMV. In this phase, the mode function is basically used, and the voted results are calculated. Based on this result, the time complexity of IMV is equal to O(i). The IMV algorithm, while being computationally lightweight, plays a crucial role in enhancing the generalization performance of the model.

### 6.5. Ablation Studies

To evaluate the individual and combined contributions of the features used in the proposed method, an ablation study was conducted. This study systematically analyzed the impact of removing specific feature groups, including statistical moments and MelPat features, on the overall classification performance. The results of this analysis are summarized in Table 8.

This ablation study highlights the critical role of both MelPat and statistical moment features in achieving a high classification accuracy. The full feature set, combining MelPat and statistical moments, yielded the highest accuracy of 97.71%. When statistical moments were excluded, the accuracy dropped to 92.53%, indicating that MelPat, while effective, relies on statistical moments to enhance its discriminative power. Conversely, using only statistical moments resulted in a significant accuracy reduction to 72.36%, demonstrating that these features alone are insufficient for robust classification. Among the statistical moments, entropy-based features were particularly influential, as their removal caused a noticeable decrease in accuracy to 69.82%, underscoring their importance in distinguishing between sleep-deprived and control groups. These results affirm that the combination of MelPat and statistical moments is essential for the model’s high performance, with each feature group contributing unique and complementary information.

### 6.6. Highlights

The key aspects of the proposed models are presented below.



Advantages:

The proposed multi-level MelPat-based new classification model has shown very high classification success. The highest classification accuracy was calculated as 97.71% on the P2 channel.The developed model has linear time complexity. Therefore, it offers an alternative solution to approaches that are popularly used today, such as deep learning.In the model, Chi2 and NCA algorithms were combined, and the power of both feature selectors was utilized jointly. In this context, the classification results obtained demonstrate the power of these combined methods.




Limitations:

There is a very limited number of datasets available in the literature. More EEG datasets are needed for testing to observe the performance of the developed model.




Future works:

Detection of sleep deprivation is a highly important topic. It has areas of use in many different disciplines. Therefore, when access to new datasets is provided, we plan to verify the performance of the developed model.In the near future, we plan to collect a new EEG dataset for sleep deprivation.Related to the subject, we aim to develop new feature extraction algorithms and achieve high classification accuracy on different datasets.


## 7. Conclusions

In this research, a new feature extraction approach has been developed by applying feature engineering. With this method, called MelPat, which is inspired by the chemical graph of the melatonin hormone, EEG signals associated with sleep deprivation have been classified. A multi-level feature extraction approach was used in the model, and feature extraction was performed from each sub-band of the EEG signal. Additionally, two feature selection algorithms were used, thereby selecting the most significant features and increasing the classification performance of the algorithm.

The developed model was tested on an open access EEG dataset and achieved 97.71% classification accuracy. The method proposed in this research has linear time complexity. Therefore, it is a cost-effective approach. It has the potential to be an alternative to deep learning approaches, which are particularly popular today.

## Figures and Tables

**Figure 1 diagnostics-15-00379-f001:**
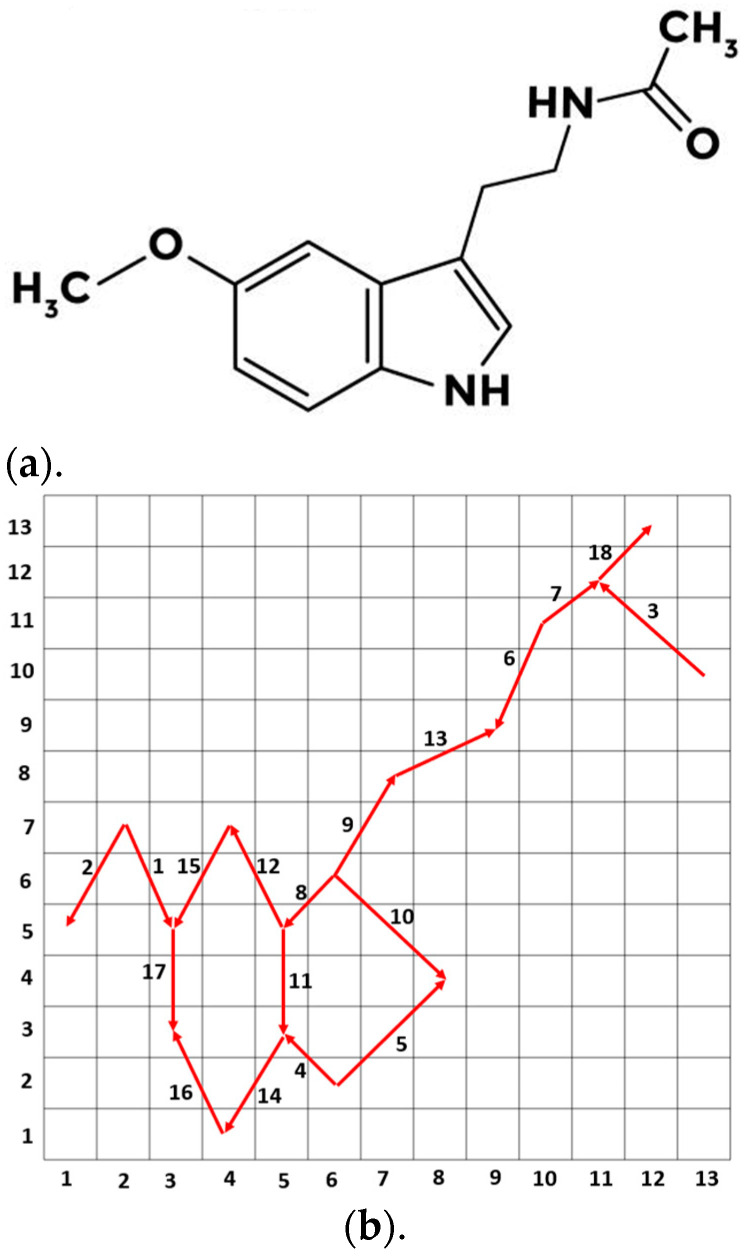
The proposed MelPat feature extractor. (**a**) Chemical graph of melatonin hormone. (**b**) Created directed graph using mMelatonin hormone (MelPat).

**Figure 2 diagnostics-15-00379-f002:**
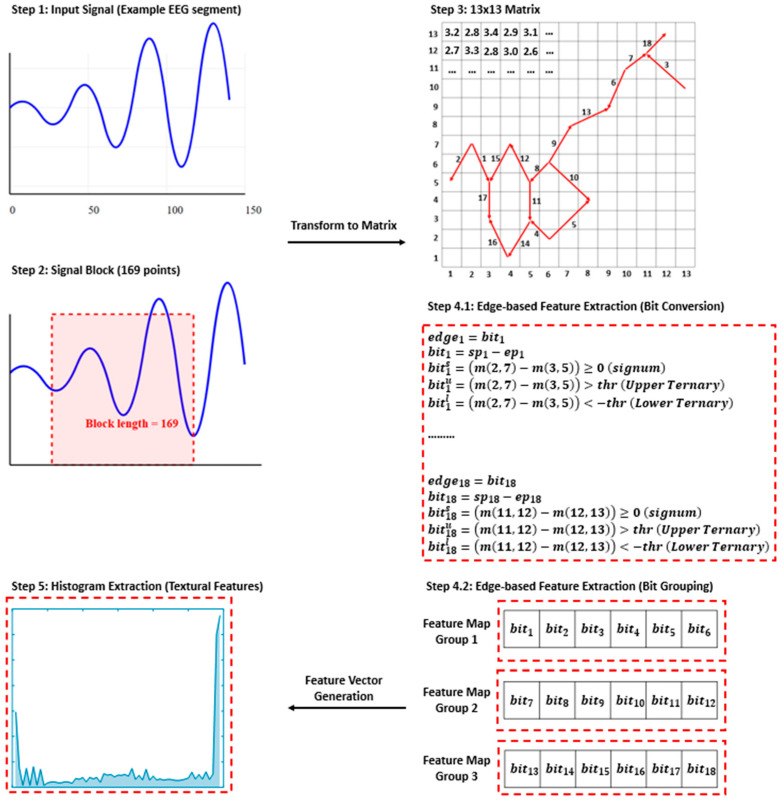
MelPat feature generator steps.

**Figure 3 diagnostics-15-00379-f003:**
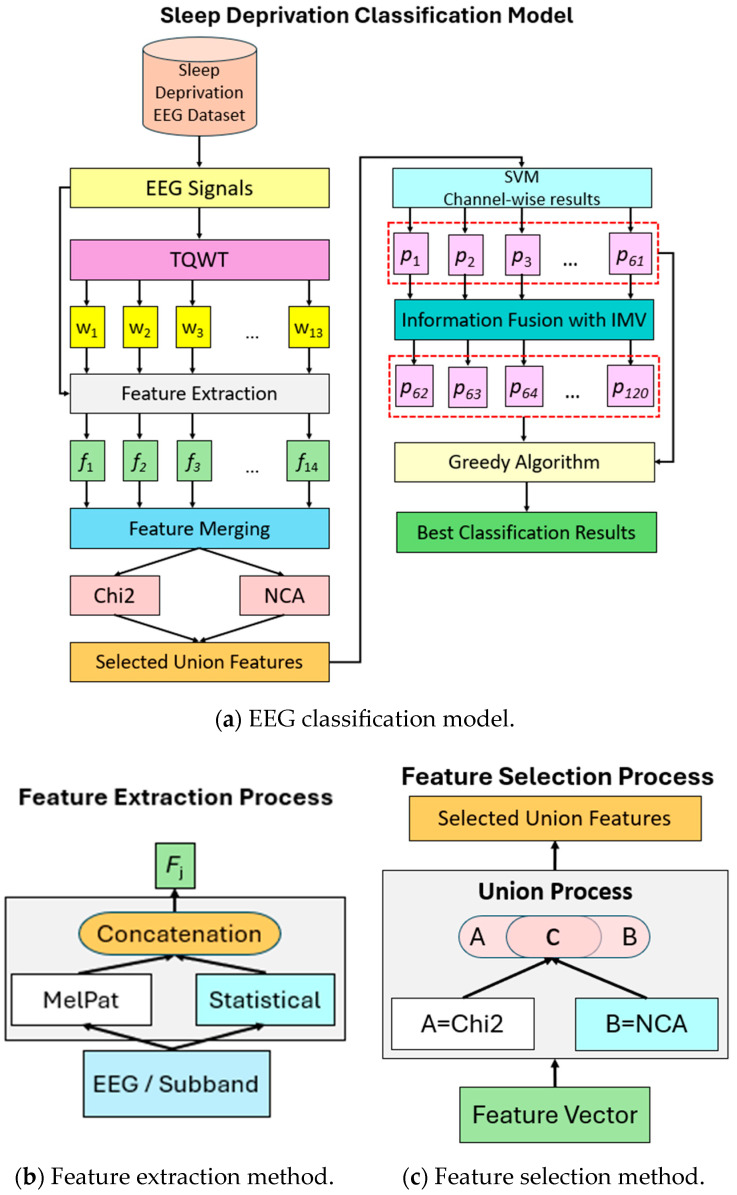
Machine learning-based classification of sleep deprivation.

**Figure 4 diagnostics-15-00379-f004:**
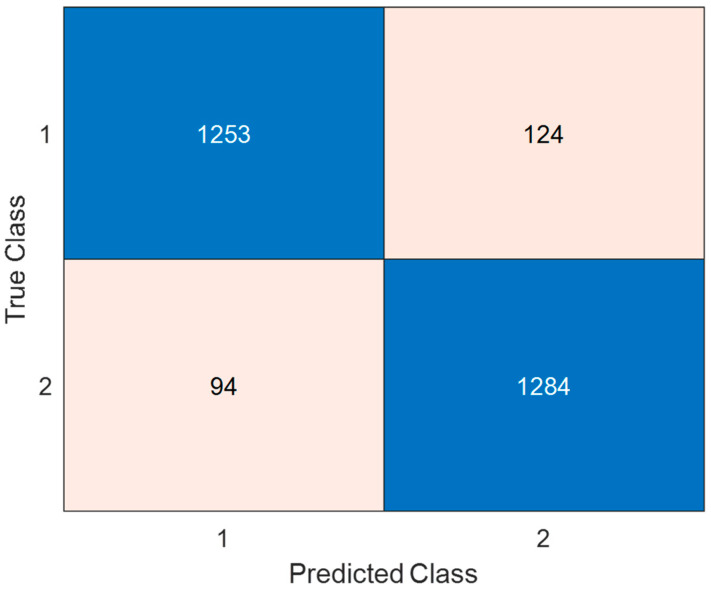
Confusion matrix for best channel-wise result (P2 channel). 1: NS = normal sleep, 2: SD = sleep deprivation.

**Figure 5 diagnostics-15-00379-f005:**
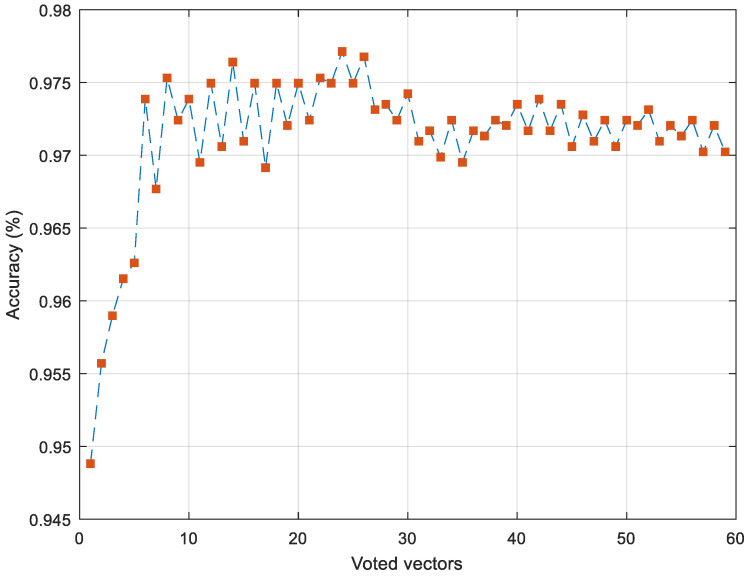
The results of voted prediction vectors using the MelPat model.

**Figure 6 diagnostics-15-00379-f006:**
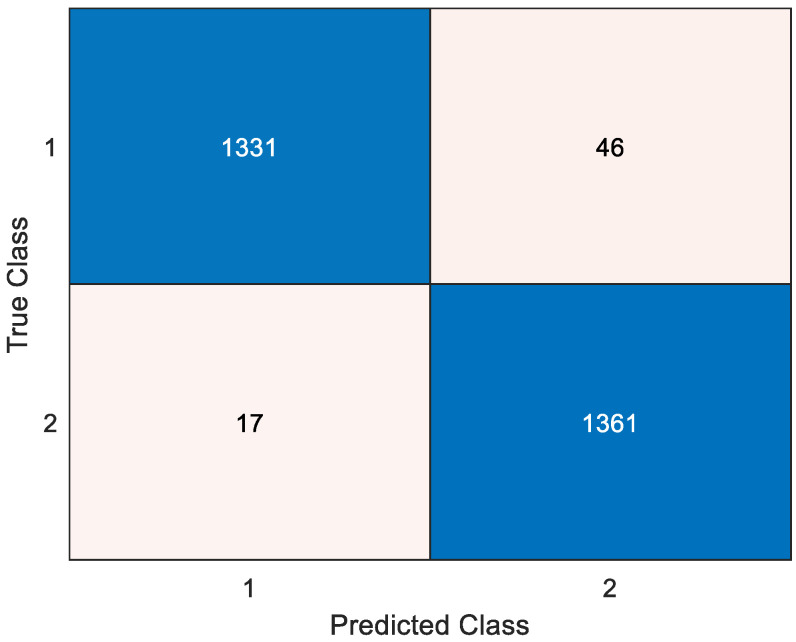
Confusion matrix for best voted prediction vector result. 1: NS = normal sleep, 2: SD = sleep deprivation.

**Figure 7 diagnostics-15-00379-f007:**
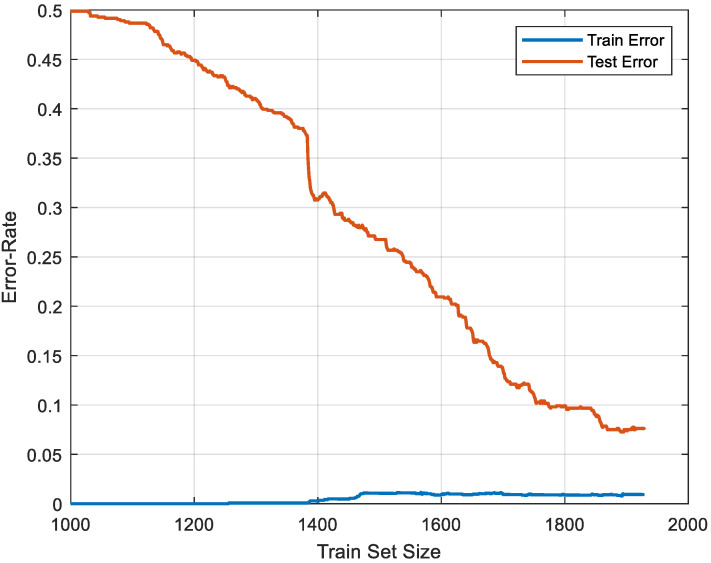
Learning curve for P2 channel using 70:30 hold-out CV.

**Figure 8 diagnostics-15-00379-f008:**
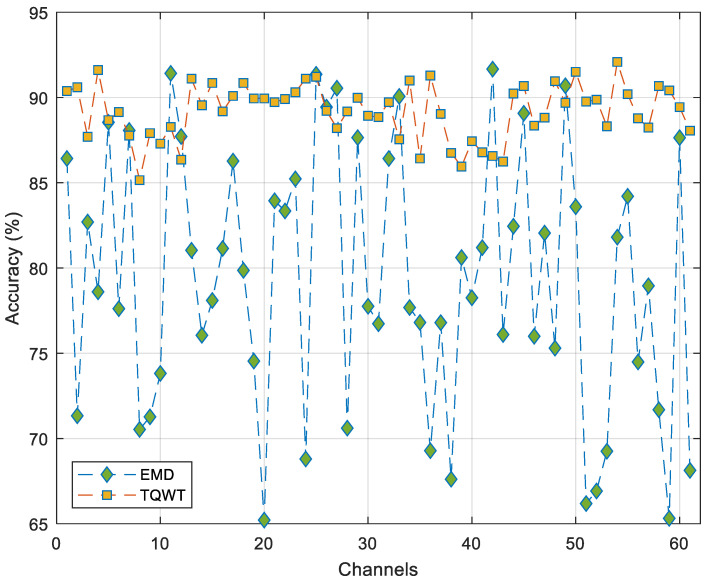
Channel-wise accuracy changes in EMD and TQWT signal decomposition methods.

**Figure 9 diagnostics-15-00379-f009:**
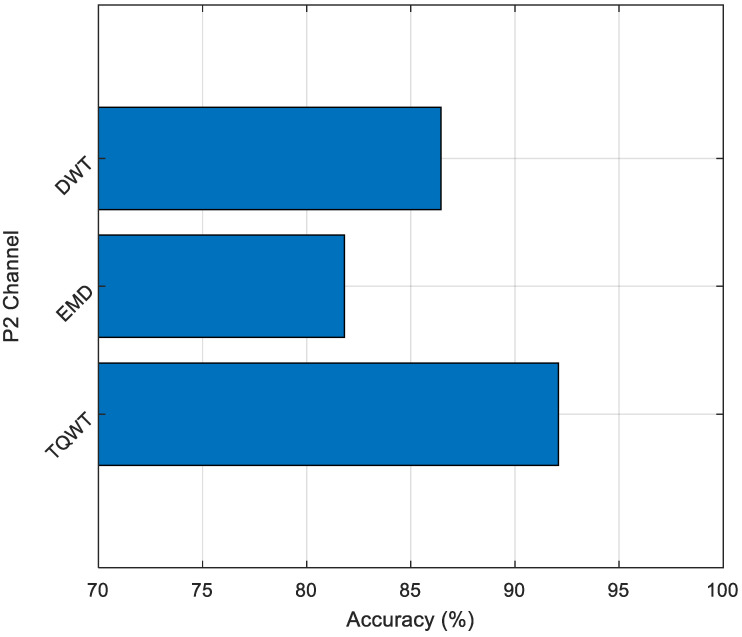
Classification accuracies for P2 channel.

**Figure 10 diagnostics-15-00379-f010:**
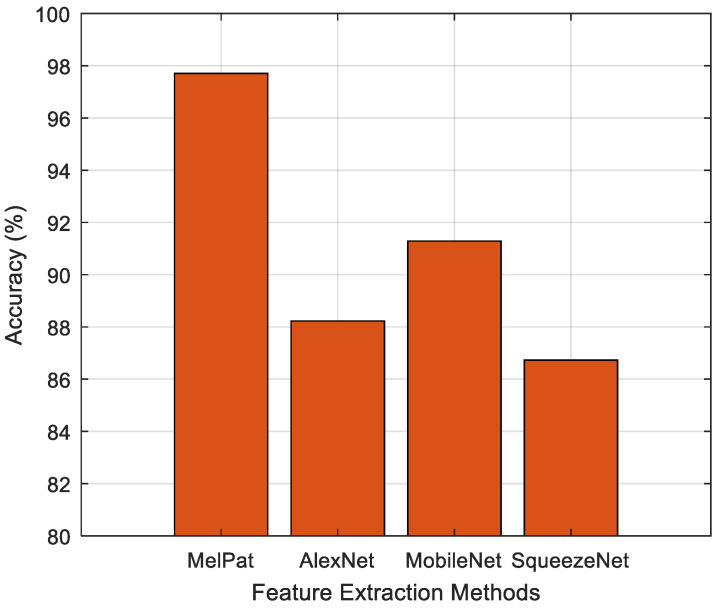
Comparison with deep feature extraction methods.

**Figure 11 diagnostics-15-00379-f011:**
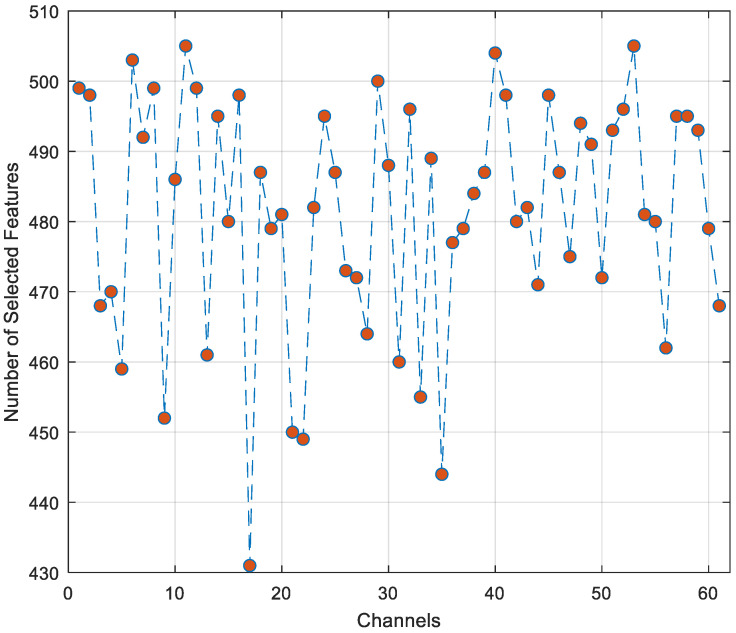
Number of selected features for each channel.

**Figure 12 diagnostics-15-00379-f012:**
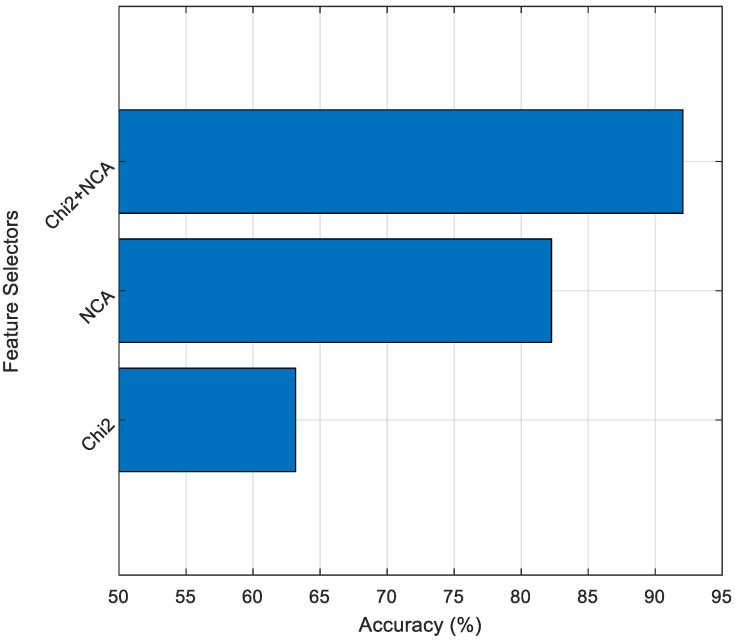
Accuracy values according to feature selection algorithms.

**Figure 13 diagnostics-15-00379-f013:**
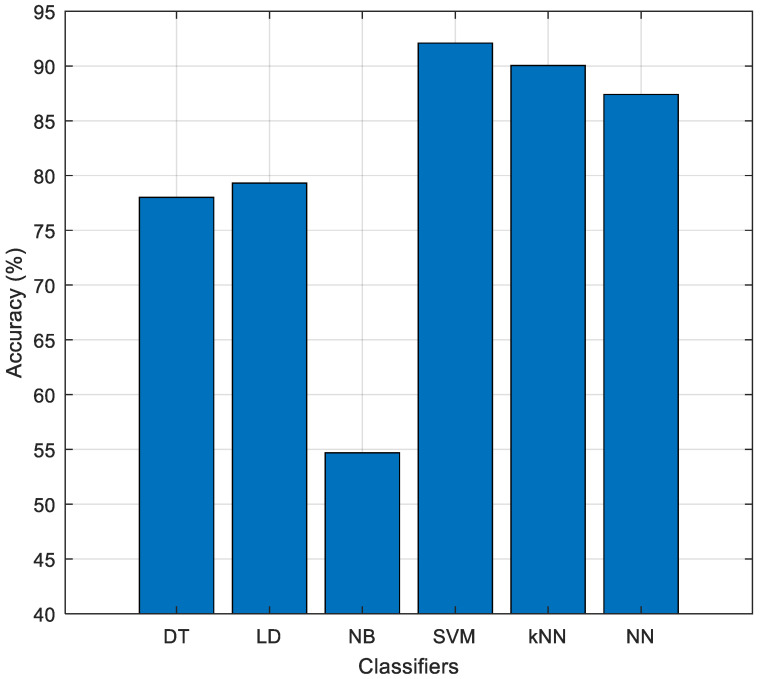
Performance values of the tested classification algorithms.

**Table 1 diagnostics-15-00379-t001:** State-of-the-art sleep deprivation detection methods.

**Author (s) and Year**	**Dataset**	**Method (s)**	**Result(s) (%)**	**Key Points**
Wang et al. 2016 [25]	8 subjects, 62 channels	Signal band decomposition, Power spectral density, linear dynamical system (LDS), kNN, SVM, and discriminative graph regularized extreme learning machine (GELM)	Acc. = 83.57	- Measuring sleep quality - Small sample size - 80:20 hold-out validation - Feature selection issues
Masri et al. 2024 [26]	10 subjects, 19 channels	Artifact and noise removing, Welch’s approach, power spectral density, kNN, SVM, ANN, and RF	Acc. = 99.7	- Sleep deprivation classification - Small sample size - 80:20 hold-out validation
Tao et al. 2020 [27]	16 subjects, 62 channels	Differential entropy (DE) features, correlation connectivity and eye movement features, and LSTM	For 6-fold CV Acc. = 86.86 For LOSO CV Acc. = 82.03	- Emotion recognition under sleep deprivation - Two validation method (6-fold and LOSO CV) - Small sample size - High computational complexity
Wall et al. 2021 [28]	24 subjects	Fourier transform, maximum intensity of the power spectrum, standard deviation of some parameters, and ANOVA	Acc. = 77.0 Spe. = 77.0 Sen. = 76.0	- Driving behavior under sleep deprivation - Small sample size - 5-fold CV - Disparity between training and test set
Lee et al. 2009 [29]	25 subjects, 1 channel	Spectral analysis and SVM	Acc. = 80	- Single channel limitation - Feature extraction dependency
Libourel et al. 2015 [30]	10 subjects (rat), 4 channels	Fast Fourier transform, normalization, Bayesian algorithm, and Mann–Whitney U-test	Spe. = 92.0	- Real-time detection of sleep and wakefulness states in rats - Limited testing across species - Dependence on initial conditions

Acc: accuracy, Sen: sensitivity, Spe: specificity, LSTM: Long Short-Term Memory, kNN: k-nearest neighbor, SVM: Support Vector Machine, ANN: Artificial Neural Network, RF: Random Forest, LOSO: Leave One Subject Out, CV: cross-validation.

**Table 2 diagnostics-15-00379-t002:** Time complexity calculation of the developed MelPat method.

Algorithm Step	Complexity	Description
Signal segmentation	O(n)	Dividing signal into blocks of length 169
Matrix transformation	O(1)	Creating fixed-size 13 × 13 matrix
Edge extraction (18 edges)	O(1)	Operations on fixed number (18) of edges
Kernel functions (sgn, ut, lt)	O(1)	Fixed operation for each edge
Feature map generation	O(1)	Fixed operation for groups of 6
Histogram extraction	O(n)	Creating histogram from feature maps
Total Complexity	O(n)	n: signal length

**Table 3 diagnostics-15-00379-t003:** Transition table of the proposed method.

Step	Input	Output
Segmentation	EEG dataset	15 s EEG segments
TQWT	EEG segments	13-wavelet bands
Feature extraction (Melpat and statistical moment)	EEG segment and 13 wavelet bands	Feature vectors of length 8624
Feature selection (Chi2 and NCA)	Feature vector of length 8624	Selected feature vector of length n≤512
Classification (SVM)	Selected feature vector	61 prediction vectors (for 61 channels)
IMV	61 predicted vectors	59 voted vectors
Greedy	120 predicted vector (=61 + 59)	1 best predicted vector

**Table 4 diagnostics-15-00379-t004:** Classification results (%) for channels of the MelPat model.

No	Ch.	Acc.	No	Ch.	Acc.	No	Ch.	Acc.
1	Fp1	90.38	21	TP7	89.73	41	FC4	86.79
2	AF3	90.60	22	TP9	89.91	42	FC6	86.57
3	AF7	87.70	23	Pz	90.31	43	FT8	86.24
4	Fz	91.62	24	P1	91.11	44	C2	90.24
5	F1	88.68	25	P3	91.22	45	C4	90.67
6	F3	89.15	26	P5	89.22	46	C6	88.35
7	F5	87.77	27	P7	88.20	47	T8	88.82
8	F7	85.15	28	PO3	89.18	48	CPz	90.96
9	FC1	87.91	29	PO7	89.98	49	CP2	89.69
10	FC3	87.30	30	Oz	88.93	50	CP4	91.51
11	FC5	88.28	31	O1	88.86	51	CP6	89.76
12	FT7	86.35	32	Fpz	89.73	52	TP8	89.87
13	Cz	91.11	33	Fp2	87.55	53	TP10	88.31
14	C1	89.55	34	AF4	91.0	54	**P2**	**92.09**
15	C3	90.85	35	AF8	86.42	55	P4	90.20
16	C5	89.18	36	F2	91.29	56	P6	88.78
17	T7	90.09	37	F4	89.04	57	P8	88.24
18	CP1	90.85	38	F6	86.75	58	POz	90.67
19	CP3	89.95	39	F8	85.95	59	PO4	90.42
20	CP5	89.95	40	FC2	87.44	60	PO8	89.44
						61	O2	88.06

**Table 5 diagnostics-15-00379-t005:** Performance metric values for P2 channel.

Performance Metric	Value (%)
Acc.	92.09
Sen.	92.09
Spe.	92.09
Pre.	92.11
F1Scr.	92.10

**Table 6 diagnostics-15-00379-t006:** Performance metric values for voted result.

Performance Metric	Value (%)
Acc.	97.71
Sen.	97.71
Spe.	97.71
Pre.	97.73
F1Scr.	97.72

**Table 7 diagnostics-15-00379-t007:** Computational complexity of the proposed method.

Step	Description	Computational Complexity
Signal Segmentation	Dividing EEG signals into 15 s non-overlapping segments	O(n)
Signal Decomposition (TQWT)	Decomposing signals into 13 sub-bands	O(nlogn)
Feature Extraction (MelPat)	Extracting features from raw signals and sub-bands	O(n)
Feature selection (Chi2 and NCA)	Informative feature selection from feature vector	O(d)+O(d)=O(2d)
Classification (SVM)	Training and testing SVM with selected features	O(s)
IMV	Combining channel-wise results using iterative voting	O(i)
Total	=O(nlogn+2d+s+i)	

**Table 8 diagnostics-15-00379-t008:** Classification accuracy after incremental removal of feature groups.

Feature Set	Accuracy (%)
Full feature set (MelPat + moments)	97.71
MelPat only	92.53
Statistical moments (include entropy features)	72.36
Statistical moments only (without entropy features)	69.82

## Data Availability

Data is available at https://doi.org/10.18112/openneuro.ds004902.v1.0.4 (accessed on 8 December 2024).

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
