# Peer review of "Melatonin Pattern: A New Method for Machine Learning-Based Classification of Sleep Deprivation"

_diagnostics, 2025, doi:10.3390/diagnostics15030379_

Round 1
Reviewer 1 Report
Comments and Suggestions for Authors
The article submitted for review concerns the use of machine learning to classify EEG signals in sleep deprivation research. The article addresses an important topic, as sleep disorders are one of the diseases of civilisation in the modern world. The authors propose a study based on EEG signals. A novelty proposed in the paper is the use of a method based on the directed graph of the chemical pattern of the hormone melatonin for feature extraction. The paper also uses other advanced methods such as TQWT, NCA, Chi2, SVM or IMV. The results obtained are interesting, achieving classification accuracy on a publicly available database of more than 97%. The paper is clearly and carefully written.
1. How was the MalPat feature extraction method found to have a ‘linear time complexity’ property, and how does this relate to the computational complexity of other feature extraction methods? What if this graph is drawn differently? The chemical formula of melatonin is the same, but different versions of the graph can be found in the literature.
2. Expressions 1, 2, 3 use the notations: sp, ep. In what sense is it start and end point - does it correspond to (x,y) coordinates in x-y plane?
3. In the pseudocode description of the MelPat method, values are used, e.g. the length of the vector is 576, or the value 168 in line 01, 02 of the pseudocode. How were such values selected?
4. It seems to me that there is a lack of a figure with a vector of data to illustrate the operation of the algorithm and the creation of the histogram, and it is not clear how to select the start point, end point, whether they are always identical, etc.
5. Why were the values of the parameters Q, r and J for the TQWT transformation chosen and not others?
6. The signal decomposition results using TQWT were compared with the EMD decomposition method. However, wouldn't it be better to see a wavelet transform here? However, EMD decomposition is of a different nature to WT or TQWT. A direct comparison between WT and TQWT would give an answer as to how much better TQWT is in determining the relevant features of the signal under analysis.
Reviewer 2 Report
Comments and Suggestions for Authors
The article presents a novel feature extraction method, Melatonin Pattern (MelPat), inspired by the melatonin hormone, for classifying sleep deprivation using EEG signals. The key innovations include the introduction of MelPat, which leverages a graph-based approach for feature extraction, and the use of TQWT for signal decomposition. The model combines NCA and Chi2 for feature selection, and employs SVM for classification, achieving a high accuracy of 97.71%. The IMV algorithm is used to generalize results across multiple EEG channels. The study demonstrates the effectiveness of MelPat in sleep deprivation detection, offering a lightweight alternative to deep learning models with linear time complexity. The study has its own significance yet I would like to seek further clarifications regarding the following major issues I have identified in the article.
1. The MelPat feature extraction method, while novel, lacks a thorough comparison with other state-of-the-art feature extraction techniques in the EEG domain. The authors should conduct a comparative analysis with methods like wavelet transforms, entropy-based features, or deep learning-based feature extraction to better justify the superiority of MelPat.
2. The paper introduces MelPat but does not provide insights into how these features correlate with sleep deprivation from a biological perspective. The authors should include a note on the interpretability of the MelPat features.
3. The article does not address potential overfitting, especially given the high dimensionality of the feature vector (8624 features) and the relatively small dataset. The authors should include cross-validation results, learning curves, or other techniques to demonstrate that the model generalizes well to unseen data.
4. The parameters (Q=2, r=4, J=12) for the Tunable Q-Wavelet Transform are specified, but the rationale behind these choices and the potential impact of varying these parameters are not thoroughly explored. A sensitivity analysis or parameter optimization study should be included to justify the selected values.
5. The iterative majority voting (IMV) algorithm is used to generalize results, but its computational complexity and scalability are not discussed; an analysis of the computational cost and scalability to larger datasets should be included.
6. The statistical moments used for feature extraction are standard, and their contribution to the model's performance is not explicitly analyzed; a feature importance analysis or ablation study could highlight the contribution of each feature type.
7. The comparison with deep learning models is mentioned, but no empirical comparison is provided; a direct comparison with deep learning baselines would better position MelPat as a lightweight alternative.
Round 2
Reviewer 2 Report
Comments and Suggestions for Authors
The authors have made significant efforts to address my concerns. I find the revised article suitable and recommend it for prospective publication. Best of luck to the authors.